DATA RELEASE

# Whole genome assembly and annotation of the King Angelfish (*Holacanthus passer*) gives insight into the evolution of marine fishes of the Tropical Eastern Pacific

Remy Gatins[1,2,3,*], Carlos F. Arias[2,4,5], Carlos Sánchez[6], Giacomo Bernardi[1] and Luis F. De León[2]

1 Department of Ecology and Evolutionary Biology, University of California Santa Cruz, Santa Cruz, CA, USA
2 Department of Biology, University of Massachusetts, Boston, MA, USA
3 Marine and Environmental Sciences, Northeastern University, Boston, MA, USA
4 Smithsonian Tropical Research Institute, Panamá, Panama
5 Data Science Lab, Office of the Chief Information Officer, Smithsonian Institution, Washington, DC, USA
6 Departamento de Ciencias Marinas y Costeras, Universidad Autónoma de Baja California Sur, CP 23080, La Paz, Baja California Sur, México

## ABSTRACT

*Holacanthus* angelfishes are some of the most iconic marine fishes of the Tropical Eastern Pacific (TEP). However, very limited genomic resources currently exist for the genus. In this study we: (i) assembled and annotated the nuclear genome of the King Angelfish (*Holacanthus passer*), and (ii) examined the demographic history of *H. passer* in the TEP. We generated 43.8 Gb of ONT and 97.3 Gb Illumina reads representing 75× and 167× coverage, respectively. The final genome assembly size was 583 Mb with a contig N50 of 5.7 Mb, which captured 97.5% of the complete Actinoterygii Benchmarking Universal Single-Copy Orthologs (BUSCOs). Repetitive elements accounted for 5.09% of the genome, and 33,889 protein-coding genes were predicted, of which 22,984 were functionally annotated. Our demographic analysis suggests that population expansions of *H. passer* occurred prior to the last glacial maximum (LGM) and were more likely shaped by events associated with the closure of the Isthmus of Panama. This result is surprising, given that most rapid population expansions in both freshwater and marine organisms have been reported to occur globally after the LGM. Overall, this annotated genome assembly provides a novel molecular resource to study the evolution of *Holacanthus* angelfishes, while facilitating research into local adaptation, speciation, and introgression in marine fishes.

**Subjects** Genetics and Genomics, Marine Biology, Evolutionary Biology

**Submitted:** 20 October 2023

* Corresponding author. E-mail: remygatinsa@gmail.com

Preprint submitted at https://doi.org/10.1101/2023.11.08.566026

## INTRODUCTION

The King angelfish, *Holacanthus passer*, is one of the most iconic fish species of the Tropical Eastern Pacific (TEP) (Figure 1). Its distribution ranges from the Northern Gulf of California (Sea of Cortez) to Peru, including the Revillagigedos, Cocos, Malpelo, and the Galápagos Islands [1, 2] (Figure 1C). Due to its conspicuous coloration, the King angelfish has become a target for the aquarium trade [2], with individuals costing between $150 and $900 (at the time of publication), while individuals of the sister species, *H. clarionensis*, endemic to the

**Figure 1.** Pictures of an adult male (A) and a juvenile (B) King angelfish, *Holacanthus passer*. Males can be identified by their white pelvic fin. The blue outline on the map (C) shows the range distribution of *H. passer* across the Tropical Eastern Pacific. The red star indicates where the sample used for the genome assembly was collected from. Photo credits: Remy Gatins.

Revillagigedos, have sold for up to $15,000. *Holacanthus passer* is currently protected under the conservation regulation in Mexico (Norma Official Mexicana) [2], but is identified as having a stable population under the IUCN red list [3]. *Holacanthus* angelfishes are protogynous sequential hermaphrodites, changing sex from female to male as they grow. They exhibit sexual dimorphism (pelvic fin coloration) (Figure 1A) [4], and can partition their habitat by sex and size classes [5]. They are important sponge feeders and herbivores but have also been observed feeding on fish feces in the water column [2, 5] and interacting as fish cleaners [6]. Additionally, their social organization can vary from solitary individuals to harems [4].

The genus *Holacanthus* is an interesting model system for assessing the drivers of diversification in marine fishes. Although it contains seven species only, it presents a complex history of diversification, which includes three modes of speciation: allopatric, peripatric, and sympatric [7, 8]. Following the closure of the Isthmus of Panama around 3.2 to 2.8 Mya [9], two clades of *Holacanthus* were separated in the Atlantic and Pacific Oceans. These so-called geminate species [10] diverged allopatrically approximately 1.7 to 1.4 Mya [7, 8, 11], along with about 40 other marine fishes (Jordan 1908; Thacker 2017) and many invertebrates [12, 13]. Within each ocean basin, additional *Holacanthus* species diverged approximately 1.5 Mya. The Tropical Eastern Pacific (TEP) clade, which consists of *H. passer*, *H. limbaughi*, and *H. clarionensis* is thought to have diverged via peripatry. In contrast, the Tropical Western Atlantic (TWA) clade, comprised by *H. bermudensis* and *H. ciliaris*, is thought to have diverged in sympatry [7, 8]. The last two *Holacanthus* species, *H. tricolor* and *H. africanus*, are considered sister taxon of the TEP-TWA clade, as well as the most ancestral *Holacanthus* taxon.

The increased accessibility of novel genomic tools has led to a rapid proliferation of whole-genome assemblies for non-model species. Recent genome assembly studies have

used of a combination of short and accurate (~99%) Illumina data with long, but less accurate, single-reads (~95%) generated by Oxford Nanopore (ONT) or PacBio sequencing [14–18]. This, hybrid assembly approach can deliver real-time targeted sequencing, while improving genome assembly contiguity and completeness [14–16, 19]. Here, we use this approach with the goal of facilitating the study of the history of diversification in *Holacanthus* and the evolutionary dynamics associated with the closure of the Isthmus of Panama in the TEP. Specifically, we use *de novo* genome sequence data to: (i) deliver a high-quality whole genome assembly of the King Angelfish, *Holacanthus passer*; and (ii) examined the demographic history of *H. passer* in the TEP.

## MAIN CONTENT

### Context

#### *Genome assembly*

The final assembled and polished genome of *Holacanthus passer* yielded a total size of ~583 Mb gathered in 486 contigs, with the largest contig at 17 Mb and a contig N50 of 5.7 Mb (Table 1). The 486 sequence fragments that make up the assembly contain zero gaps and are therefore described as contigs instead of scaffolds throughout the text. The final assembly was slightly larger than the initial ~579 Mb estimated by GenomeScope (Figure 2A) as well as the initial 581 Mb assembly before the polishing iterations. Kraken identified approximately 100 kb of potential contaminants, none of which were identified using Blobtools (Figure 2B) and were thus retained in the assembly. Detailed assembly statistics after the first initial assembly and consecutive polishing rounds can be found in Table 1. The number of contigs remained at 486 throughout the assembly. After four iterations of polishing using ONT and Illumina reads, BUSCO completeness improved from 82.4% to 97.5% for the Actinopterygii dataset ($n$ = 4,584) and 90.1% to 95.4% in the Eukaryota dataset ($n$ = 303). The largest completeness increase (10.6%) in the BUSCO Actinopterygii dataset occurred after the first ONT polishing iteration, while in the Eukaryota dataset the highest increase (2.3%) occurred after the first ONT polishing and the second Illumina polishing (Table 1). Additionally, the N50 contig length increased from 5.6 to 5.7 Mb after polishing. These results indicate that polishing with both ONT and Illumina reads greatly improved the assembly, by correcting assembly bases, fixing misassemblies, and filling assembly gaps. Moreover, contiguity did not improve after the initial assembly carried out with the Wtdbg2 assembler using long ONT reads. This suggests that the assembler and initial input reads play an important role in how contiguous the assembled genome is, while multiple polishing iterations can improve the accuracy of the assembly. The King angelfish genome assembly presented here is comparable in quality to other recently published fish genomes. When comparing this genome with the only other available genome assembly of the Pomacanthidae family, *H. passer* showed a slightly smaller genome (580 Mb) than *Centropyge vrolikii* (696.5 Mb) [18] (Table 2, Figure 3B). Additionally, our King angelfish genome resulted in a much more contiguous assembly (*H. passer*: 450 contigs; *C. vrolikii*: 30,500 scaffolds) and showed a significant lower number of gaps throughout (*H. passer*: 0 gaps; *C. vrolikii*: 30,486 gaps). In spite of *H.* passer having a smaller N50 (5.7 Mb) than *C. vrolikii* (9 Mb) (Table 2), *H. passer* showed a slightly higher number of complete orthologous matches in BUSCO than *C. vrolikii* (Figure 3). When compared with numerous other recently published chromosome level fish genomes, *H. passer* showed comparable, if not higher, BUSCO scores, despite not being a chromosome level assembly (Figure 3). In general, our

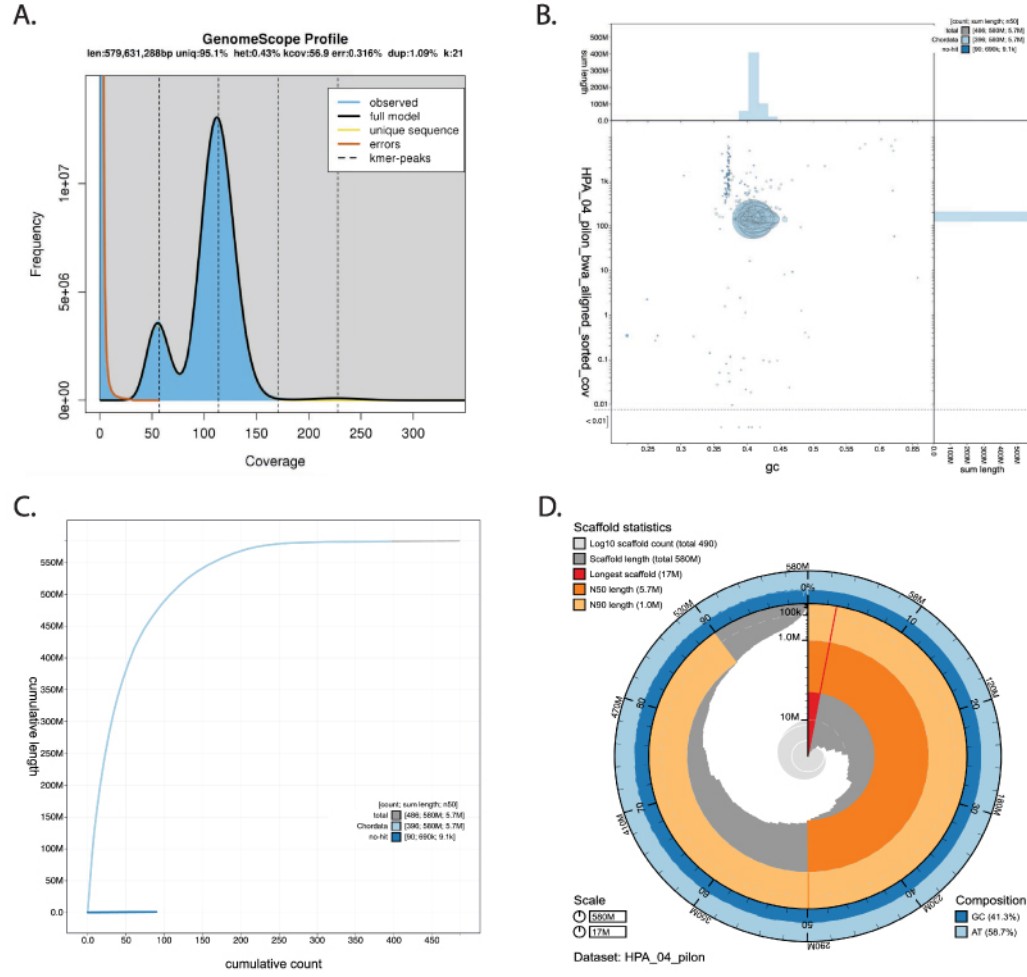

**Figure 2.** Genome assembly summary statistics of *Holacanthus passer*. Visualizations were generated using GenomeScope VX (A) and Blobtoolkit Viewer (B–D). (A) Histogram of the 21 k-mer distribution of Illumina short reads. The highest frequency of k-mer coverage was seen around 110X (excluding k-mers with low coverage). (B) Blob plot showing the distribution of assembly scaffolds based on GC proportion and coverage. Circles are colored by phylum and circle size is relative to the number of sequence length. (C) Cumulative assembly plot showing curves of subsets of scaffolds assigned to each phylum relative to the overall assembly. (D) Snail plot summary of the genome assembly statistics. The outermost ring of the entire plot represents the full length of the genome assembly of *Holacanthus passer* (583,528,366 bp). The dark blue and light blue shaded area represents the GC and AT content across the entire genome, respectively. The second ring shows the percentage of total contigs (second black line), with the light orange shade indicating the N90 (1,000,532 bp) and dark orange shade the N50 (5,708,022 bp). The dark grey bars represent the scaffolds and are organized from largest (shown in red) to smallest. The radius of the circle indicates the size of each scaffold and helps visualize the percentage of large versus small scaffolds.

assembly is highly contiguous with zero gaps, which could result in less fragmented genes. Overall, this *H. passer* assembly will serve as a high-quality genomic reference assembly for the Pomacanthidae family. This assembly also illustrates how N50 values do not always correlate with the best BUSCO scores as outlined in Jauhal and Newcomb [20].

### Genome annotation
RepeatMasker estimated that 5.09% of the genome consisted of repetitive sequences, primarily LINEs (0.85%), LTR elements (0.31%), DNA transposons (1.36%), and simple

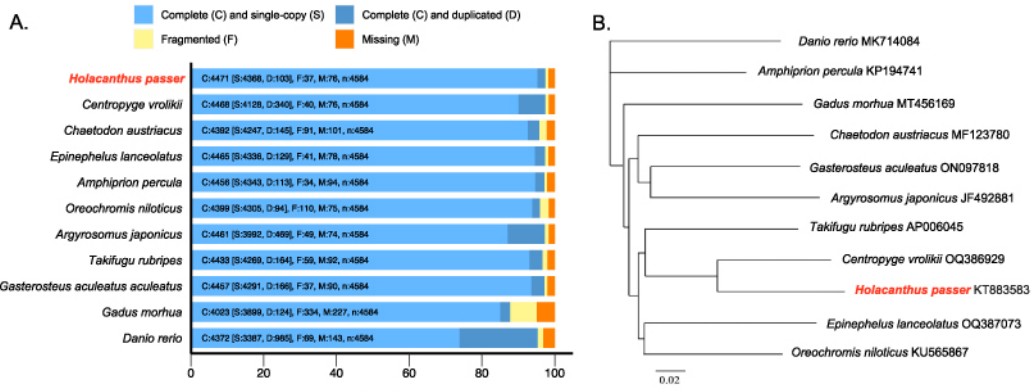

**Figure 3.** Comparative analysis of fish genome assemblies. (A) BUSCO completeness of the *Holacanthus passer* genome assembly (red) assessed by the 4,584 orthologous actynopterygii (odb9) dataset compared to other available fish genome assemblies. *Centropyge vrolikii* is the closest available fish genome within the Pomacanthidae family. (B) The phylogeny depicting the relationships among these fish genomes using the COI marker can be seen in panel B.

**Table 1.** Step-by-step genome assembly and annotation statistics of the King angelfish (*Holacanthus passer*).

| | Nanopore | | | Nanopore + Illumina | |
|---|---|---|---|---|---|
| **Genome assembly** | **Wtdbg2** | **Wtdbg2 + 1× Racon** | **Wtdbg2 + 2× Racon** | **Wtdbg2 + 2× Racon + 1× Pilon** | **Wtdbg2 + 2× Racon + 2× Pilon** |
| Total assembly size of contigs (bp) | 581,422,425 | 583,574,933 | 583,552,491 | 583,601,337 | 583,528,366 |
| Number of contigs | 486 | 486 | 486 | 486 | 486 |
| N50 contig length (bp) | 5,681,869 | 5,707,473 | 5,709,778 | 5,708,674 | 5,708,022 |
| N90 contig length (bp) | 997,074 | 1,000,168 | 1,000,597 | 1,000,715 | 1,000,532 |
| Longest contig (bp) | 17,088,287 | 17,147,963 | 17,147,963 | 17,150,647 | 17,148,928 |
| GC % | | | | | 41.27 |
| **Actinopterygii** | | | | | |
| Complete BUSCOs | 3,779 (82.4%) | 4,263 (93%) | 4,296 (93.7%) | 4,468 (97.5%) | 4,471 (97.5%) |
| Complete and single-copy BUSCOs | 3,674 (80.1%) | 4,133 (90.2%) | 4,163 (90.8%) | 4,364 (95.2%) | 4,368 (95.3%) |
| Complete and duplicated BUSCOs | 105 (2.3%) | 130 (2.8%) | 133 (2.90%) | 104 (2.3%) | 103 (2.2%) |
| Fragmented BUSCOs | 374 (8.2%) | 176 (3.8%) | 155 (3.40%) | 38 (0.8%) | 37 (0.8%) |
| Missing BUSCOs | 431 (9.4%) | 145 (3.2%) | 133 (2.9%) | 78 (1.7%) | 76 (1.7%) |
| **Eukaryota** | | | | | |
| Complete BUSCOs | 273 (90.1%) | 280 (92.4%) | 280 (92.4%) | 282 (93.10%) | 289 (95.4%) |
| Complete and single-copy BUSCOs | 267 (88.1%) | 270 (89.10%) | 270 (89.10%) | 267 (88.1%) | 274 (90.4%) |
| Complete and duplicated BUSCOs | 6 (2.0%) | 10 (3.3%) | 10 (3.3%) | 15 (5%) | 15 (5%) |
| Fragmented BUSCOs | 4 (1.3%) | 3 (1%) | 4 (1.3%) | 2 (0.7%) | 2 (0.7%) |
| Missing BUSCOs | 26 (8.6%) | 20 (6.60%) | 19 (6.3%) | 19 (6.2%) | 12 (3.9%) |
| **Annotation** | | | | | |
| Number of protein-coding genes | | | | | 33,793 |
| Mean gene length (bp) | | | | | 10,807 |
| Number of CDSs | | | | | 388,693 |
| Longest gene (bp) | | | | | 315,502 |
| Functionally annotated | | | | | 22,992 |

repeats (2.14%) (Table 3). Repeat content was nearly identical to that estimated by GenomeScope (4.9%). GeMoMa identified 33,793 gene models and 388,693 CDSs, where 67.8% (22,992) of the gene models had a functional annotation (Table 1). The number of

**Table 2.** Comparison summary statistics for 11 selected fish genome assemblies, including *Holacanthus passer* from this study.

| | *Holacanthus passer* | *Centropyge vrolikii* | *Chaetodon austriacus* | *Epinephelus lanceolatus* | *Amphiprion percula* | *Oreochromis niloticus* | *Argyrosomus japonicus* | *Takifugu rubripes* | *Gasterosteus aculeatus* | *Gadus morhua* | *Danio rerio* |
|---|---|---|---|---|---|---|---|---|---|---|---|
| Common name | King angelfish | Pearlscale pygmy angelfish | Blacktail butterflyfish | Giant grouper | Orange clownfish | Nile tilapia | Japanese meagre | Fugu | Three-spined stickleback | Atlantic cod | Zebrafish |
| Family | Pomacanthidae | Pomacanthidae | Chaetodonidae | Serranidae | Pomacentridae | Cichlidae | Sciaenidae | Tetraodontidae | Gasterosteidae | Gadidae | Cyprinidae |
| **Platform** | | | | | | | | | | | |
| Shotgun | x | | x | | | | | | | | x |
| Illumina (paired-end) | x | | x | x | | x | x | | | | x |
| Mate-Pairs | | x | x | | | | | | | | |
| 10 × Genomics | | | | | | | | x | | x | |
| Nanopore | x | | | | | | | | | x | |
| PacBio | | | | | x | x | x | x | x | x | |
| Hi-C | | | | x | x | | x | x | x | x | |
| Chicago | | x | | | | | | | | | |
| BioNano | | | | | | | | | | x | |
| **Total length (Mb)** | 583.5 | 696.5 | 712.2 | 1,087.4 | 909.0 | 1,005.7 | 792.0 | 384.1 | 471.9 | 684.3 | 1,679.2 |
| % GC | 41.27 | 41.76 | 42.48 | 41.26 | 39.53 | 40.73 | 41.25 | 45.67 | 44.66 | 45.69 | 36.6 |
| **Scaffolds** | | | | | | | | | | | |
| Number | 486 | 30,500 | 13,441 | 4,200 | 366 | 2,459 | 1,984 | 127 | 2,911 | 1,126 | 1,922 |
| N50 length (Mb) | 5.7 | 9 | 0.17 | 46.2 | 38.4 | 38.8 | 13.1 | 16.7 | 20.4 | 27.4 | 52.2 |
| Longest (Mb) | 17.1 | 31 | 2 | 57.7 | 46.1 | 87.6 | 30.3 | 29.2 | 34.2 | 41.8 | 78.1 |
| Ns (Kb) | 0.0 | 11,709.3 | 48,772.7 | 39,254.8 | 32.4 | 55.0 | 0.0 | 3,688.5 | 3,574.2 | 12.6 | 4,693.6 |
| Gaps | 0 | 30,486 | 105,028 | 23,415 | 682 | 551 | 0 | 402 | 3,125 | 126 | 20,258 |
| **Chromosomes** | | | | 24 | 24 | 23 | 24 | 22 | 22 | 23 | 25 |
| % Masked * | 5.09 | 15.94 | | 2.61 | 2.92 | 5.4 | | 10.26 | | 10.17 | 57.77 |
| GenBank assembly accession | | | | GCA_005281545.1 | GCA_003047355.2 | GCA_001858045.3 | GCA_015710095.1 | GCA_901000725.2 | GCA_016920845.1 | GCA_010882105.1 | GCA_000002035.4 |
| RefSeq assembly accession | | | | GCF_005281545.1 | GCF_002776465.1 | GCF_001858045.2 | | GCF_901000725.2 | GCF_016920845.1 | GCF_902167405.1 | GCF_000002035.6 |
| Reference | This study | Fernandez-Silva et al. 2018 | DiBattista et al. 2018 | Zhou et al. 2019 | Lehman et al. 2018 | Conte et al. 2017 | Zhao et al. 2021 | | | Kirubakaran et al. 2020 | |

* indicates percent masked reported using RepeatMasker program.



**Table 3.** Summary output of repetitive elements of *H. passer* predicted by RepeatMasker v. 2.9.0+. The query species was assumed to be *Danio rerio*.

| Sequences: | 486 | | |
|---|---|---|---|
| Total length: | 583528366 bp (583528366 bp excl N/X-runs) | | |
| GC level: | 41.27% | | |
| Bases masked: | 29714081 bp (5.09 %) | | |
| | Number of elements* | Length occupied (bp) | Percentage of sequence |
| **Retroelements** | 32,172 | 6,939,127 | 1.19% |
| SINEs: | 1,265 | 127,915 | 0.02% |
| Penelope | 303 | 35,535 | 0.01% |
| LINEs: | 19,022 | 4,977,480 | 0.85% |
| CRE/SLACS | 0 | 0 | 0.00% |
| L2/CR1/Rex | 13,025 | 3,278,888 | 0.56% |
| R1/LOA/Jockey | 644 | 120,329 | 0.02% |
| R2/R4/NeSL | 299 | 122,053 | 0.02% |
| RTE/Bov-B | 1,556 | 536,242 | 0.09% |
| L1/CIN4 | 2,571 | 723,359 | 0.12% |
| LTR elements: | 11,885 | 1,833,732 | 0.31% |
| BEL/Pao | 1,085 | 311,809 | 0.05% |
| Ty1/Copia | 25 | 16,958 | 0.00% |
| Gypsy/DIRS1 | 6,190 | 1,075,740 | 0.18% |
| Retroviral | 2,370 | 223,106 | 0.04% |
| **DNA transposons** | 67,101 | 7,958,272 | 1.36% |
| Hobo-Activator | 24,022 | 2,283,863 | 0.39% |
| Tc1-IS630-Pogo | 10,113 | 2,729,978 | 0.47% |
| En-Spm | 0 | 0 | 0.00% |
| MuDR-IS905 | 0 | 0 | 0.00% |
| PiggyBac | 217 | 31,373 | 0.01% |
| Tourist/Harbinger | 2,025 | 231,198 | 0.04% |
| Other (Mirage, P-element, Transib) | 1,538 | 262,794 | 0.05% |
| Rolling-circles | 421 | 48,093 | 0.01% |
| Unclassified: | 269 | 71,601 | 0.01% |
| Total interspersed repeats: | | 14,969,000 | 2.57% |
| Small RNA: | 1,676 | 161,165 | 0.03% |
| Satellites: | 961 | 80,911 | 0.01% |
| Simple repeats: | 303,686 | 12,479,070 | 2.14% |
| Low complexity: | 38,530 | 2,144,093 | 0.37% |

coding sequences identified for *H. passer* was within the range of those found in other closely related fish species genomes (see [21]; assembled and annotated fish genomes, visited April 28, 2021).

### Demographic history of H. passer

The demographic history analysis of *H. passer* showed two extreme scenarios (Figure 4). When considering a faster mutation rate (μ) of $10^{-8}$, the population showed a slow expansion ~300 Kya, with a small population decline occurring ~70 Kya, followed by a second rapid expansion 30 Kya, reaching a maximum effective population size of ~300,000 individuals (Figure 4A). When using a slower mutation rate of $10^{-9}$, the population showed an initial expansion around 2.8 Mya, with a small decline ~600 Kya, and the subsequent rapid expansion 300 Kya, reaching a maximum effective population size of ~2,800,000 individuals (Figure 4B).

Considering the slower mutation rate scenario, an effective population size in the order of millions of individuals for *H. passer* seems plausible. In particular, because this species occupies a vast available habitat compared to its sister species *H. limbaughi* whose effective



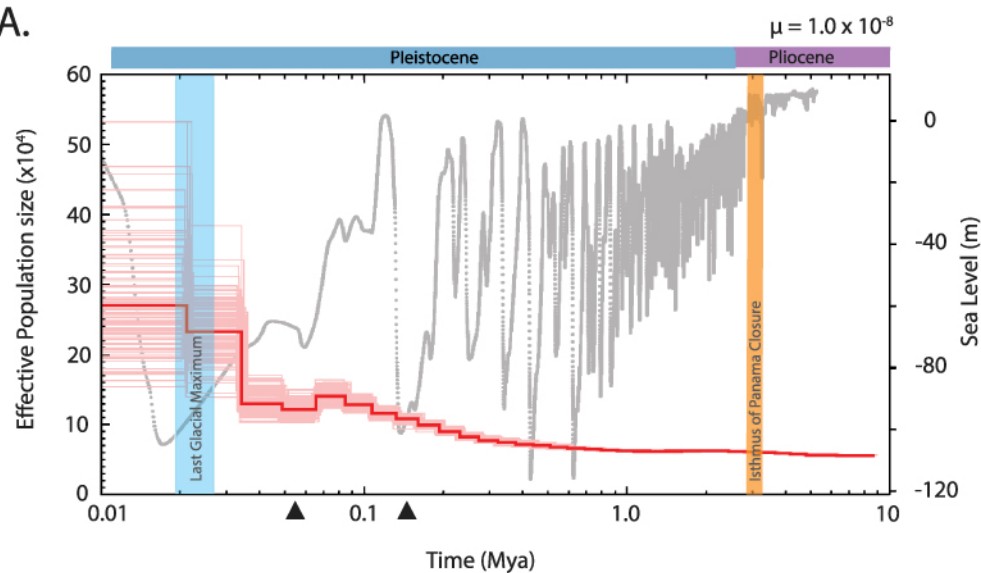

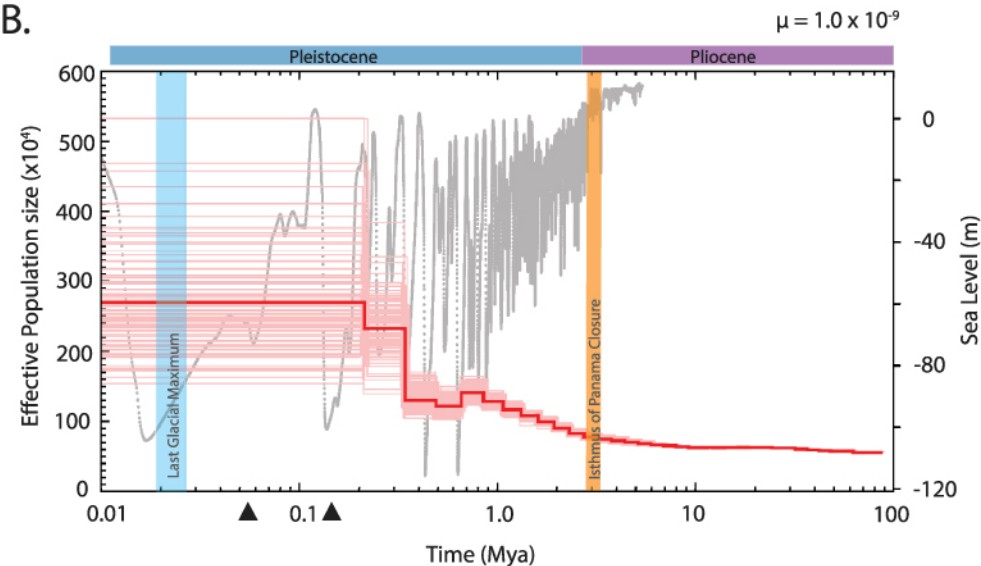

**Figure 4.** Genome-wide demographic history in *Holacanthus passer*. PSMC analysis showing the demographic history (red line) of *H. passer* using a generation time of 5 years and a mutation rate (μ) of 10–8 (A) and 10–9 (B). Global sea level model fluctuations over the past 5 million years are shown in the background (grey) (data from [22]). Vertical blue bars refer to the last glacial maximum (LGM) period (~19–26.5 kya) and the orange bar represents the closure of the Isthmus of Panama (~3.2–2.8 Mya). Triangles represent marine population expansion events previously recorded in the Tropical Eastern Pacific (see text).

population size is estimated to be ~60,000 individuals [23]. *H. limbaughi* is endemic to Clipperton Island and occupies a fraction of the distribution of *H. passer*, which is found across the entire TEP coastline. However, considering the higher mutation rate scenario may also seem likely when considering the first rapid population expansion occurred much after the closure of the Isthmus of Panama once oceanographic conditions in the TEP became more suitable.

*H. passer* was previously estimated to have diverged from its geminate Atlantic species (*H. ciliaris*) between 1.7 and 1.4 Mya [7, 8], based on a molecular clock calibrated according to the closure of the Isthmus of Panama around 3.1 to 3.5 Mya [11]. However, recent studies suggest the closure of the Isthmus of Panama might have happened more recently, around 2.8 Mya [9]. Therefore, the genetic divergence between *Holacanthus* geminates could be more recent than previously believed.

After the closure of the Isthmus, oceanographic conditions in the TEP varied drastically following sea level changes due to multiple glaciation periods in the Pleistocene [24, 25], likely leading to important demographic consequences [26]. Most rapid population expansions in both freshwater [27, 28] and marine organisms [29] have been reported to occur globally after the last glacial maximum (LGM) that took place from 26.5 to 19 Kya [30]. However, only a few species have reported population expansions prior to the LGM [29]. On the contrary, in the TEP, most studies that have assessed the demographic history of marine organisms have found population expansions that precede the LGM [31–34] and few reporting population expansions in the last 20 Kya [34, 35]. For instance, the goby, *Elacatinus puncticulatus*, and the clingfish, *Gobieosox adustus*, experienced a population expansion around 170–130 Kya and 200–150 Kya, respectively [31, 33]. While another reef fish, *Anisotremus interruptus*, experienced an expansion in its continental populations after the LGM (~5 kya). Interestingly, *A. interruptus* populations from the oceanic islands of Revillagigedos and the Galapagos Archipelago showed earlier expansions at around 55 kya [34]. Yet, all demographic history studies in the TEP to date are based on single mitochondrial markers.

To the best of our knowledge, our study is the first to assess the demographic history of a marine fish in the TEP using genome-wide nuclear DNA. Our results support previous findings of marine population expansions in the TEP occurring prior to the LGM [31–34]. This pattern is consistent with our analyses using both slow and fast mutation rates for *H. passer*, which showed population expansions beyond 30 Kya. Overall, drops in sea level are likely to decrease the available marine habitat, potentially restricting gene flow between populations, and resulting in population bottlenecks. This was particularly prominent in areas where shallow marine habitats (<60 m) are abundant, such as the Western Atlantic, Western Pacific, and Eastern Indian Ocean [26]. Map projections of the TEP during the LGM show relatively small differences of the exposed landmasses at low sea level (−60m) compared to present day [26]. This suggests that glaciation sea level drops might not have changed the overall topology and gene flow in the TEP in contrast to other ocean basins. Overall, although our demographic estimates varied considerable with our choice of mutation rate, our results are generally consistent with previous studies indicating that population expansions of marine fishes in the TEP may have preceded the LGM [31]. Furthermore, this also suggests that the demographic history in *H. passer* was likely shaped by historical events associated with the closure of the Isthmus of Panama, rather than by the more recent LGM.

## Methods
### Sample collection and DNA extraction
Fin and gill clips were collected from 13 individuals of *Holacanthus passer* in La Paz, Baja California Sur, Mexico (Figure 1). Collections were made with pole spears while SCUBA diving, abiding by IACUC protocols. Tissue samples were immediately placed in 95%

ethanol and stored at −20 °C. DNA was extracted using a DNeasy Blood and Tissue kit according to manufacturer's protocol (Qiagen). DNA quality and concentration of the 13 samples were assessed using a Nanodrop 2000c and Qubit 4.0 Fluorometer. The sample with the highest quality was further evaluated on an Agilent 2200 TapeStation DNA ScreenTape to check for high molecular weight. The sample chosen for the genome assembly of *Holacanthus passer* had a final DNA concentration of 205 ng/µl, a 260/280 and 260/230 ratio of 2.02 and 2.26, respectively, and an average fragment length of 38 kb (Figure 5A). This sample came from an adult *H. passer* female with a total length size of 20.4 cm. Before beginning with our library prep, DNA was transferred from AE buffer to EB to remove traces of EDTA, as recommended by Nanopore library prep, using a 3× KAPA Pure Bead clean up (Roche Molecular Systems). DNA was then eluted in 90 µl of EB, reaching a final concentration of 128 ng/µl. This sample was sequenced using ONT and Illumina (HiSeq4000; 150 bp paired-end, RRID:SCR_016386) sequencing.

### *Whole-genome library construction and sequencing*

Four individual ONT libraries were prepared with 1.5 µg of DNA using the SQK-LSK109 library prep protocol according to manufacturer's protocol (Oxford Nanopore Technologies, Oxford, UK). DNA was first sheared using the Covaris g-TUBE, following the manufacturer's protocol for 10 kb fragments to improve output yield (Figure 5B). One ONT library was prepared without DNA shearing to target longer fragments; however, N50 only increased by about 1 kb, while the output yield decreased between half to a third. Thus, we opted to continue to shear DNA for the remaining libraries. Each library was sequenced on a R9.4 flow cell using the MinION DNA sequencer (RRID:SCR_017985). Maximum run time ranged between 48 to 72 hours. Raw data was basecalled separately using Guppy 3.3 basecaller on a GPU-based high-performance computer cluster server of the University of Massachusetts Boston. A total of 43.8 Gb ($N_{50}$: 6,626 bp, longest read: 474,205 bp) were generated on the Oxford Nanopore MinION device. Individual statistics can be found in the GitHub repository [36].

The Illumina library was prepared with 250 ng of unsheared DNA using the Kapa Hyperplus Library Preparation Kit with only one third of the volume reactions as described in the manufacturer's protocol (Kapa Biosystems, Wilmington, MA). The total fragmentation volume was 16.66 µl and was incubated at 37 °C for 7:45 min. The incubation parameters were previously optimized to target fragments of ~500 bp. Post-ligation purification was done using a 0.8× KAPA Pure bead cleanup. Library amplification was carried out with a total PCR reaction volume of 16.6 µl for 8 PCR thermal cycles. Finally, we did a double size-selection post-amplification cleanup with SPRIselect beads using a 0.56× upper and 0.72× lower selection ratio (Beckman Coulter, Inc) (Figure 5C). The final Illumina library was sequenced in a pool of three individuals with a HiSeq4000 (150 bp paired-end) (Novogene Corporation Inc.), which generated a total of 97.3 Gb sequencing data with an average cleaned read of 149 bp.

GenomeScope (RRID:SCR_017014) [37] was used to estimate genome size, repeat content, and heterozygosity across all k-mers ($k$ = 21) previously detected using Jellyfish v2.2.10 (RRID:SCR_005491) [38], to help choose parameters for downstream analysis. Using only raw Illumina data, the genome size of *H. passer* was estimated to have a length of 579 Mb with approximately 95.1% of unique content and a heterozygosity level of 0.43% (Figure 2A). Additionally, k-mers with 110× coverage showed the highest frequency.

**A.**

**B.**

| Peak | | Size [bp] | Conc. [pg/µl] | Molarity [pmol/l] |
|---|---|---|---|---|
| 1 | ◄ | 35 | 125.00 | 5,411.3 |
| 2 | | 1,302 | 12.85 | 15.0 |
| 3 | | 1,620 | 77.51 | 72.5 |
| 4 | | 2,963 | 2,105.51 | 1,076.6 |
| 5 | | 3,940 | 448.96 | 172.7 |
| 6 | | 4,357 | 1,935.78 | 673.1 |
| 7 | | 8,977 | 78.49 | 13.2 |
| 8 | ▶ | 10,380 | 75.00 | 10.9 |

**C.**

| Peak | | Size [bp] | Conc. [pg/µl] | Molarity [pmol/l] |
|---|---|---|---|---|
| 1 | ◄ | 35 | 125.00 | 5,411.3 |
| 2 | | 197 | 2.21 | 17.0 |
| 3 | | 502 | 2,357.89 | 7,113.0 |
| 4 | | 511 | 238.74 | 708.3 |
| 5 | | 524 | 1,766.48 | 5,103.1 |
| 6 | | 1,514 | 8.16 | 8.2 |
| 7 | | 2,202 | 16.22 | 11.2 |
| 8 | | 3,114 | 19.31 | 9.4 |
| 9 | | 5,486 | 18.30 | 5.1 |
| 10 | | 6,401 | 15.32 | 3.6 |
| 11 | | 7,381 | 12.49 | 2.6 |
| 12 | | 8,280 | 10.81 | 2.0 |
| 13 | ▶ | 10,380 | 75.00 | 10.9 |

**Figure 5.** *Holacanthus passer* genomic DNA profile used for Nanopore Sequencing. (A) TapeStation analysis using a Genomic DNA ScreenTape (Agilent Technologies, Inc 2017) of DNA sample used pre-fragmentation. Peak molecular weight was found to be at 31,831 bp with a calibrated concentration of 19.6 ng/µl. Between 250 and 60,000 bp, a region representing 84% of the sequences, the average size was 18,931 bp with a concentration of 23.5 ng/µl. (B,C) Bioanalyzer 2100 profile and statistics using a High Sensitivity DNA Assay (Agilent Technologies, Inc 2009) of genomic DNA post sheared with Covaris g-TUBE following manufacturers protocol for 10 kb fragments (B) and after Kapa Hyperplus library prep followed by a double size-selection cleanup with SPRIselect beads (0.56× and 0.72×) (C).

Considering a genome size of 579 Mb, the output of 43.8 Gb of ONT and 97.3 Gb of Illumina reads represented a total of 75× and 167× coverage respectively, based on the size of our final genome assembly.

### Genome assembly

Long reads obtained from the ONT were concatenated into one large fastq file and trimmed with Porechop v. 0.2.3 (RRID:SCR_016967) [39]. Nanofilt v. 2.5.0 (RRID:SCR_016966) [40]) was used to create two different filtered data sets to help the contiguity of the final assembly. Our top five longest reads ranged from 176 kb to 474 kb with an average quality score (Q) of 3.9. Thus, the first data set was filtered to keep sequences with a minimum Q score of 3 and sequence length of 1,000 bp as it resulted in the most contiguous assembly (Nanofilt parameters -q 3; -l 1000). For the second data set we increased the Q score to 5 and it was explicitly used for downstream assembly polishing (-q 5 and -l 500). The former sequences were assembled using Wtdbg2 v2.5 (RRID:SCR_017225) [41], setting a minimum sequence length of 1,000 bp (-L 1000). To improve the draft assembly, two rounds of consensus correction were performed using the -q 5 filtered ONT reads, by mapping reads to the draft genome with Minimap2 v. 2.17 [42] and polishing with Racon v. 1.4.7 [43].

Short accurate Illumina reads were used to further polish the ONT genome. Raw sequences were adapter-trimmed with Trimmomatic v. 0.39 (RRID:SCR_011848) [44] and quality checked before and after trimming using FastQC v 0.11.8 (RRID:SCR_014583) [45]. Two rounds of polishing were carried out by mapping the trimmed short reads to the assembly using BWA v 0.7.17 (RRID:SCR_010910) [46], sorted and indexed with Samtools v 1.9 (RRID:SCR_002105) [47], and consensus corrected using Pilon v 1.23 (RRID:SCR_014731) [48].

Finally, given that the DNA used for the genome assembly was extracted from gill tissue, which could be exposed to microorganisms, the final assembly was screened for sequences of bacteria, viruses, and plasmids using Kraken 2.0.9 (RRID:SCR_005484) [49] and Blobtools2 [50]. Any contaminants found and in accordance with both programs were removed from the final assembly. Genome completeness was assessed using Benchmarking Universal Single-Copy Orthologs (BUSCO v3.0.2) (RRID:SCR_015008) [51, 52] by comparing the *H. passer* genome to the Actinopterygii (*n* = 4,584) and Eukaryota (*n* = 303) ortholog gene datasets. Assembly statistics and BUSCO completeness were assessed after the initial draft assembly, and subsequently, after each polishing iteration (Table 1). The complete flow chart of the full genome assembly pipeline is shown in Figure 6. In order to take this assembly one step further into a chromosome-level genome, future research should build upon this assembly and incorporate proximity ligation technology, such as Hi-C or Omni-C (e.g., [53]).

### Genome assembly comparison

Ten genome assemblies were selected from Genbank to compare with our genome assembly [17, 18, 54–58]. These assemblies represent a broad range of genomic technologies, ranging from close to distantly related species (Table 2, Figure 3). All genomes were downloaded from NCBI and genome statistics and BUSCO completeness were assessed using the same methods described above. To conceptualize the relationships between each species, we plotted their phylogenetic relationships based on the mitochondrial cytochrome oxidase 1 gene (COI). COI loci were obtained from GenBank, and aligned using Geneious (v10.2.6) [59]. The alignment was then used to reconstruct phylogenetic trees based on Neighbor-Joining approaches with the APE (Analyses of Phylogenetics and Evolution) R-package [60].

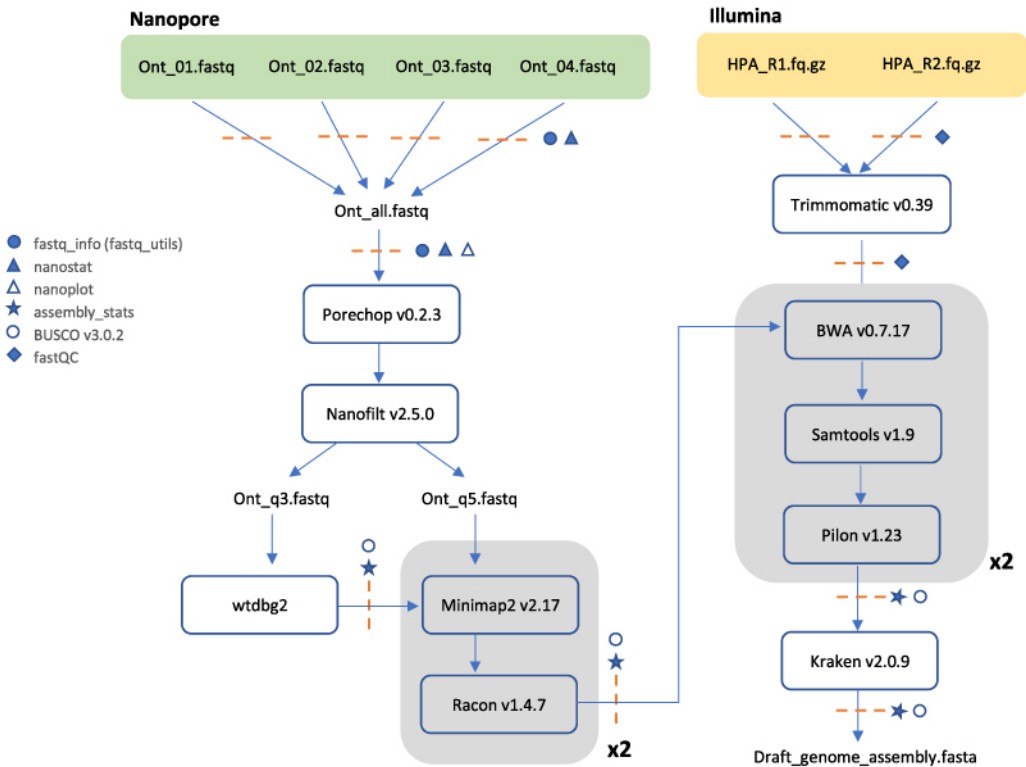

**Figure 6.** Whole genome assembly pipeline using Oxford Nanopore and Illumina sequencing. Dashed orange lines indicate quality assessment checkpoints carried out during the assembly pipeline.

### Genome annotation

To annotate our genome, we used the homology-based gene prediction pipeline GeMoMa (v1.6.4, RRID:SCR_017646) [61, 62]. GeMoMa uses protein-coding genes models and intron position conservation from reference genomes to predict possible protein-coding genes in a target genome. We ran the GeMoMa pipeline using annotations from three fish species: *Amphiprion ocellaris*, *Oreocromis niloticus*, *Electrophorus electricus* (downloaded from NCBI, see Table 4). These species were selected to represent a variety of genes from close to distant high-quality fish annotations. In our case, the pipeline performed four main steps: (1) Extractor or external search, using the search algorithm tbalstn with cds parts as queries from our reference genomes, (2) Gene Model Mapper (GeMoMa), which builds gene models from the extractor results , (3) GeMoMa Annotation Filter (GAF) that filters and combines common gene predictions and (4) AnnotationFinalizer, which predicts UTRs for annotated coding sequences and generate genes and transcripts names [61]. Additionally, repetitive elements were predicted by running RepeatMasker (open-4.0.6, RRID:SCR_012954) [63] with the Teleostei database to identify repetitive elements in the genome and soft-mask the assembly. RepeatMasker.out was converted to GFF with RepeatMasker script 'rmOutToGFF3.pl'.

### Demographic history of H. passer

To infer the demographic history of *H. passer* in the TEP, a Pairwise Sequentially Markovian Coalescent (PSMC) model was used to explore temporal changes in effective population size

**Table 4.** Reference genomes and annotations used to predict gene models with the GeMoMa pipeline.

| Common name | Scientific name | RefSeq assembly | Genome and annotation release link | Download date | Annot. release |
|---|---|---|---|---|---|
| Electric Eel | *Electrophorus electricus* | fEleEle1.pri (GCF_013358815.1) | https://ftp.ncbi.nlm.nih.gov/genomes/all/annotation_releases/8005/101/GCF_013358815.1_fEleEle1.pri/ | 12/1/20 | 101 |
| Clown Anemonefish | *Amphiprion ocellaris* | AmpOce1.0 (GCF_002776465.1) | https://ftp.ncbi.nlm.nih.gov/genomes/all/annotation_releases/80972/101/GCF_002776465.1_AmpOce1.0/ | 12/1/20 | 101 |
| Nile Tilapia | *Oreochromis niloticus* | O_niloticus_UMD_NMBU (GCF_001858045.2) | https://ftp.ncbi.nlm.nih.gov/genomes/all/annotation_releases/8128/104/GCF_001858045.2_O_niloticus_UMD_NMBU/ | 12/1/20 | 104 |

based on genome-wide diploid sequence data [64]. The PSMC analysis is particularly powerful to infer demographic histories beyond 20,000 years, which fits well with the known history of the *Holacanthus* genus [7, 8]. The PSMC simulation was run with 30 iterations (-N), a maximum 2N0 coalescent time of 30 (-t), initial theta/rho ratio of 5 (-r), and the pattern parameter (-p) set to "4 + 30 × 2 + 4 +6 + 10" [64, 65]. Generation time (g) is defined as the age at which half of the individuals of the population are reproducing. Given that *H. passer* is protogynous, generation time for females is around three years, while for males it is around six years, after they transition from female to male [2, 66, 67]. Thus, we set the average generation time (-g) for *H. passer* to 5 years. Mutation rate (μ) per site per generation in fishes has been previously estimated to be between $10^{-8}$ to $10^{-9}$ mutations per site [23, 68], thus we ran two simulations to represent the potential range of the expected mutation rates.

## REUSE POTENTIAL

This study presents the first annotated genome assembly of the King Angelfish, *Holacanthus passer*. It also provides a genomic resource to improve our understanding of the evolution of *Holacanthus* angelfishes, while facilitating novel research into local adaptation, speciation, and introgression of marine fishes. In addition, this genome will improve our understanding of the evolutionary history and population dynamics of marine species in the Tropical Eastern Pacific.

## AVAILABILITY OF SOURCE CODE AND REQUIREMENTS

- Project name: *Holacanthus passer* ONT Illumina Genome Assembly
- Project home page:
  https://github.com/remygatins/Holacanthus_passer-ONT-Illumina-Genome-Assembly
- Operating system(s): Platform independent
- Programming language: Markdown
- Licence: MIT.

## DATA AVAILABILITY

The genome assembly and raw sequencing reads (Illumina and Nanopore) have been deposited into NCBI under BioProject PRJNA713824 and are linked to Biosample SAMN18269499. The GenBank accession number of the genome assembly is JAFREQ000000000.1. Genome annotation and any additional annotation files can be found in Dryad [69]. Step-by-step code to reproduce the methods can be found in GitHub [36].

## LIST OF ABBREVIATIONS

bp: base pair; BUSCO: Benchmarking Universal Single-Copy Orthologs; g: Generation time; Gb: gigabase; kb: kilobase; Kya: Thousand years ago; LGM: Last Glacial Maximum; Mya: Million years ago; ONT: Oxford Nanopore; PSMC: Pairwise Sequentially Markovian Coalescent model; TEP: Tropical Eastern Pacific; TWA: Tropical Western Atlantic; μ: Mutation rate.

## DECLARATIONS

### Ethics approval

The care and handling of all vertebrate animals used in this study was in compliance with the Institutional Animal Care and Use Committee (IACUC/BERNG-1601) of the University of California Santa Cruz and performed in accordance with the Molecular Ecology and Evolution of Fishes Laboratory.

### Consent for publication

Not applicable.

### Competing interests

The authors declare that they have no competing interests.

### Author Contributions

RG, CFA, CS, GB, and LFD designed the project. RG and CS collected the samples. RG and CFA carried out the molecular lab work and bioinformatic analyses. All authors contributed to writing and revising the manuscript.

### Funding

Molecular and computational resources were funded by the Department of Biology and the Ronald E. McNair Post-Baccalaureate Achievement Program at the University of Massachusetts Boston. RG was financially supported by the Consejo Nacional de Ciencia y Tecnología (CONACYT) and the University of California Institute for Mexico and the United States (UC-MEXUS) under the Contract No. 536570.

### Acknowledgements

We would like to thank the lab members from Proyecto de Fauna Arrecifal lab at the Universidad de Baja California Sur, La Paz (UABCS) for helping secure fieldwork logistics and sample collections. We would additionally like to thank the Umass Boston High Performance Computing team for their assistance with our computational research needs.

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
