## [Editor Report]

Editor’s AssessmentThe King Angelfish (Holacanthus passer) is a great example of a Holacanthus angelfish that are some of the most iconic marine fishes of the Tropical Eastern Pacific. However, very limited genomic resources currently exist for the genus and these authors have assembled and annotated the nuclear genome of the species, and used it examine the demographic history of the fish. Using nanopore long reads to assemble a compact 583 Mb reference with a contig N50 of 5.7 Mb, and 97.5% BUSCOs score. Scruitinising the data, the BUSCO score was high compared to the initial N50’s, providing some useful lessons learned on how to get the most out of ONT data. The analysis suggests that the demographic history in H. passer was likely shaped by historical events associated with the closure of the Isthmus of Panama, rather than by the more recent last glacial maximum. This data provides a genomic resource to improve our understanding of the evolution of Holacanthus angelfishes, and facilitating research into local adaptation, speciation, and introgression of marine fishes. In addition, this genome can help improve the understanding of the evolutionary history and population dynamics of marine species in the Tropical Eastern Pacific.

---

## [Reviewer Report]

Reviewer name and names of any other individual's who aided in reviewer Iria Fernandez SilvaDo you understand and agree to our policy of having open and named reviews, and having your review included with the published papers. (If no, please inform the editor that you cannot review this manuscript.)YesIs the language of sufficient quality?YesPlease add additional comments on language quality to clarify if needed
A "the" is missing before "clingfish" in line 171 Are all data available and do they match the descriptions in the paper? YesAdditional CommentsAre the data and metadata consistent with relevant minimum information or reporting standards? See GigaDB checklists for examples <a href="http://gigadb.org/site/guide" target="_blank">http://gigadb.org/site/guide</a>YesAdditional CommentsIs the data acquisition clear, complete and methodologically sound?YesAdditional CommentsIs there sufficient detail in the methods and data-processing steps to allow reproduction?YesAdditional CommentsIs there sufficient data validation and statistical analyses of data quality? YesAdditional CommentsIs the validation suitable for this type of data?YesAdditional CommentsIs there sufficient information for others to reuse this dataset or integrate it with other data?YesAdditional CommentsAny Additional Overall Comments to the AuthorThe genome assembly presented is of high quality, with values of accuracy and completeness in pair with chromosome level assemblies. The study is very well presented in terms on quality of the results and clarity in the presentation of methods and results. An added value is that it allows understanding how different type of data and assemblers interact in improvng the assembly quality. I also found interesting to see how contiguity and completeness are not always correlated, as this assembly has a great completeness BUSCO score in spite of not having the greatest N50 (compared with the most modern assemblies). This is possibly inherent to the type of data (ONT reads) and this information may guide researchers in making decission over future assembly projects. The demographic analysis is a nice addition to the study, the results are coherent and add information interesting to study the evolution of reef fishes and the biogeography of the TEP.  I would appeciate more detail in the captions of figure 4, particularly those of the figure 4D.RecommendationAccept

---

## [Reviewer Report]

Reviewer name and names of any other individual's who aided in reviewer Yue SongDo you understand and agree to our policy of having open and named reviews, and having your review included with the published papers. (If no, please inform the editor that you cannot review this manuscript.)YesIs the language of sufficient quality?YesPlease add additional comments on language quality to clarify if needed
Are all data available and do they match the descriptions in the paper? YesAdditional CommentsAre the data and metadata consistent with relevant minimum information or reporting standards? See GigaDB checklists for examples <a href="http://gigadb.org/site/guide" target="_blank">http://gigadb.org/site/guide</a>YesAdditional CommentsIs the data acquisition clear, complete and methodologically sound?YesAdditional CommentsIs there sufficient detail in the methods and data-processing steps to allow reproduction?YesAdditional CommentsIs there sufficient data validation and statistical analyses of data quality? YesAdditional CommentsIs the validation suitable for this type of data?YesAdditional CommentsIs there sufficient information for others to reuse this dataset or integrate it with other data?YesAdditional CommentsAny Additional Overall Comments to the AuthorThe sequencing and annotation of King Angelfish genomes is impressive and represents a significant addition to the genomic resources for marine fishes. By hybrid assembly, a high-quality genome was provided, and the relationship between historical dynamics of its population and geological events was further discussed. However, in the section on inferring the demographic history, there is no mention of how the author inferred the mutation rate of this species. In addition, the author obtained 486 contigs throughout the assembly using ONT data combined with short reads. Is it possible to further assemble these contigs into chromosomal level? Of course, this does not indicate that it must be achieved within this manuscript, but rather suggests the inclusion of additional discussion on methods to further enhance the referential value of this genome.  Additional specific comments： (1) Line 86, I guess the author probably meant to say there were 486 contigs, right? (2) Line 294, "gene models", not "gen models" (3) Line 110-111, it is puzzled my about the numbers in parentheses. I don't quite understand what these numbers mean. I haven't seen any explanation in this MS. Did I miss something? (4) If possible, it is recommended to show the phylogenetic relationships between these species in Figure 3.RecommendationMinor Revision